# Cloning and Characterization of *Aedes aegypti* Trypsin Modulating Oostatic Factor (TMOF) Gut Receptor

**DOI:** 10.3390/biom11070934

**Published:** 2021-06-23

**Authors:** Dov Borovsky, Kato Deckers, Anne Catherine Vanhove, Maud Verstraete, Pierre Rougé, Robert G. Shatters, Charles A. Powell

**Affiliations:** 1Department of Biochemistry and Molecular Genetics, University of Colorado Anschutz School of Medicine, Aurora, CO 80045, USA; 2Zoological Institute, Katholieke Universiteit Leuven, 3000 Leuven, Belgium; katodeckers@gmail.com (K.D.); ac.vanhove@gmail.com (A.C.V.); Verstraete.maud@gmail.com (M.V.); 3UMR 152 Pharma-Dev, Faculté des Sciences Pharmaceutiques, Institut de Recherche et Développement, Université Toulouse 3, F-31062 Toulouse, France; pierre.rouge@free.fr; 4USDA ARS, Subtropical Horticultural Laboratory, 2001 Rock Road, Ft. Pierce, FL 34945, USA; robert.shatters@usda.gov; 5UF-IFAS Indian River Research and Education Center, Fort Pierce, FL 34945, USA; cpowell@ufl.edu

**Keywords:** mosquito, ABC-TMOF receptor, molecular modeling, sequencing, kinetic characterization, fluorescence microscopy, *E. coli*

## Abstract

Trypsin Modulating Oostatic Factor (TMOF) receptor was solubilized from the guts of female *Ae. Aegypti* and cross linked to His_6_-TMOF and purified by Ni affinity chromatography. SDS PAGE identified two protein bands (45 and 61 kDa). The bands were cut digested and analyzed using MS/MS identifying a protein sequence (1306 amino acids) in the genome of *Ae. aegypti*. The mRNA of the receptor was extracted, the cDNA sequenced and cloned into pTAC-MAT-2. *E. coli SbmA^−^* was transformed with the recombinant plasmid and the receptor was expressed in the inner membrane of the bacterial cell. The binding kinetics of TMOF-FITC was then followed showing that the cloned receptor exhibits high affinity to TMOF (K_D_ = 113.7 ± 18 nM ± SEM and B_max_ = 28.7 ± 1.8 pmol ± SEM). Incubation of TMOF-FITC with *E. coli* cells that express the receptor show that the receptor binds TMOF and imports it into the bacterial cells, indicating that in mosquitoes the receptor imports TMOF into the gut epithelial cells. A 3D modeling of the receptor indicates that the receptor has ATP binding sites and TMOF transport into recombinant *E. coli* cells is inhibited with ATPase inhibitors Na Arsenate and Na Azide.

## 1. Introduction

Oostatic hormones and antigonadotropins, factors that inhibit egg development, have been shown in cockroaches, decapod crustaceans and the house fly [1,2,3,4]. In mosquitoes the ovary synthesizes a factor that inhibits yolk deposition in less developed ovaries and in *Rhodnius prolixus* oostatic hormone peptide (M_r_ = 1.411 kDa) is produced by the abdominal neurosecretory organs [5,6,7,8,9]. Even though the hormone inhibits egg development the hormonal targets vary. In the house fly it was proposed that the hormone inhibits the release or synthesis of egg developmental neurosecretory hormone (EDNH), whereas in mosquitoes it was proposed that the hormone acts directly on the ovary inhibiting egg development and ecdysteroid biosynthesis [4,5,10]. Borovsky [11] showed that *Ae. aegypti* “oostatic hormone” does not act on the ovary or endocrine tissues but on the midgut cells stopping trypsin activity and blood digestion and causing inhibition of egg development in *Ae. aegypti* in *Culex quinquefasciatus, Culex, nigripalpus* and *Anopheles albimanus* [11]. Therefore, the hormone is not species specific but found in many mosquito species. As the hormone targets the midgut cells and not the ovary or the brain as was earlier proposed, the hormone was named “Trypsin Modulating Oostatic Factor” (TMOF). The hormone is synthesized as unblocked decapeptides (YDPAPPPPPP (A) and DYPAPPPPPP (B)) with similar activities but different abundances [12]. *Aedes aegypti* (*Aea*) TMOF B (DYPAPPPPPP) and its truncated analogue DYP, are cleaved from *Ae. aegypti* vitelline membrane genes (GenBank accession numbers S54556 and S54555, respectively) [13] like the growth inhibiting peptides of *Neobellieria* colloostatin that are cleaved from a collagen like precursor [14]. The gene coding for *Aea*TMOF A (*tmf*A) has not yet been identified. However, the genes for *Drosophila* TMOF and the nematodes peptide (YDPLPPPPPP, GenBank accession number CEY37D8A.21) have been reported [15,16].

Synthetic peptide analogues of TMOF exhibited biological activity when tested with adults and larval mosquitoes [12,17,18,19]. A 3D conformation of TMOF by NMR showed that TMOF has a rigid structure in solution and the polyproline portion of the molecule exhibits a left-handed alpha helix [20]. *Aea*TMOF is synthesized by the mosquito ovary 18 h after the blood meal and is secreted into the hemolymph reaching a peak at 33 h and rapidly declines to a minimum at 48 h [21]. Cytoimmunochemical studies indicate that the site of synthesis of *Aea*TMOF is the follicular epithelium of the ovary, the brain, the abdominal ganglia, and the upper gut of adult female *Ae. aegypti*. In larvae, TMOF reactive cells were detected in the prothoracic and abdominal ganglia suggesting that *Aea-*TMOF is an ovarian and neurosecretory peptide [19,21].

TMOF effect on the trypsin gene in *Neobellieria* was followed by injecting *Neo*-TMOF (10^−9^ M) into the hemolymph of these flesh-flies. Northern blot analysis showed that TMOF inhibited the translation of the late trypsin mRNA in the gut without affecting the integrity of the late trypsin message [22]. Similar results were reported when TMOF was expressed on the coat protein of engineered Tobacco Mosaic Virus and the virus was fed to larval *H. virescens*. Although the fed larvae transcribed a normal level of trypsin transcript, it was not translated [23]. Injecting TMOF into *Ae. aegypti* and *Cx. quinquefasciatus* and following the late trypsin transcript in the female gut showed that in mosquitoes similar to *Neobellieria* and *Heliothis* the trypsin message was transcribed but not translated [24].

To characterize the TMOF gut receptor, complementary decapeptides were used on *Ae. aegypti* gut membranes showing that the binding to the gut membrane is pH and temperature dependent. High and low affinity specific binding sites were found (K_d1_ = 4.6 + 0.7 × 10^−7^ M and K_d2_ = 4.43 ± 1 × 10^−6^ M, respectively). The specific TMOF receptor binding sites on female mosquito gut increased after the blood meal and they were visualized by immunochemistry staining. However, the receptor proteins were not sequenced or identified [25].

These results indicate that TMOF binds gut epithelial cells using gut specific receptor(s) stopping the translation of the late trypsin message. As TMOF has biological activity when it acts on mosquitoes and other insects [12,17,18,19,23] and can be used as a biological agent to control larval mosquitoes, it is important to sequence and characterize the TMOF receptor to be able to identify other insects with similar receptors that could be biologically controlled using TMOF-like peptides. TMOF is a polyproline rich peptide similar to bacterial antimicrobial proline rich peptides that enter bacterial cells using SbmA a dimeric inner membrane protein that exhibits similar 3D conformation with ATP Binding Cassette (ABC) [26,27,28]. We explored the possibility that mosquitoes, like plants [29], use ATP Binding Cassette (ABC) importer as *Aea*TMOF receptor to stop trypsin mRNA translation in the gut epithelial cells.

We report here for the first time, sequencing, 3D conformation model, and molecular characterization of *Aea*TMOF gut receptor.

## 2. Materials and Methods

### 2.1. Insects, Bacterial Strains, and Chemicals

Larval *Ae. aegypti* were reared at 26 °C on a diet of brewer’s yeast and lactalbumin (1:1) under 16:18 h light:dark cycle. Adults were fed on 10% sucrose or on chicken blood. Females were used 3–5 days after emergence.

*E. coli* CGSC strain7636: F^−^*D(araD-araB)567, DlacZ4787*(::rrnB-3), *l^−^*, *rph-1,D(rhaD-rhaB)568, hsdR514* and *E. coli* CGSC strain 8547: 

F^−^
*D(araD-araB)567, DlacZ4787*(::rrnB-3), *DsbmA742::kan, l^−^*, *rph-1,D(rhaD-rhaB)568, hsdR514* (http://cgsc.biology.yale.edu/StrainRpt.php) were grown in Luria-Bertani (LB) at 37 °C under aerobic conditions with the addition, when required, of the appropriate antibiotics at the following concentrations 100 μg/mL for ampicillin and 50 μg/mL for kanamycin. Valinomycin, 2,4-dinirophenol, sodium azide and sodium arsenate were purchased from Sigma-Aldrich and diluted in M9 medium.

### 2.2. Preparation of Gut Membrane Proteins

Female *Ae. aegypti* were fed a blood meal and 72 h later, after the blood was digested, guts were dissected out in a drop of saline and transferred into an Eppendorf tube containing 100 mM PBS pH 7.2 (1.0 mL) and 100 μL of Protease inhibitor cocktail (Thermo Fisher Corporation, Waltham, MA USA). The tube was centrifuged for 5 min at 5000 rpm using an Eppendorf desk centrifuge, the supernatant removed, and the pellet was thoroughly homogenized with a Teflon tip. The homogenate was recentrifuged at 5000 rpm, the supernatant removed, and the pellet extracted using a Membrane Protein Extraction Kit (Mem-PER Thermo Fisher, USA) following manufacturer instructions. After extraction, the solubilized membrane proteins were centrifuged at 15,000 rpm for 3 min and the supernatant dialyzed against 0.1 M PBS pH 7.2 (100 mL) containing 0.5% CHAPS for 3 h at 4 °C in a dialysis bag with M_r_ cutoff of 3.5 kDa. The dialysis buffer was replaced, and the dialysis was repeated one more time. After dialysis, 50 μL of protease inhibitor cocktail was added to the extracted membrane protein (0.75 mL, 3.5 mg) and the extract stored at −80 °C.

### 2.3. Cross Linking of TMOF to Its Soluble Gut Receptor 

His_6_-TMOF was synthesized using standard automated solid phase peptide techniques [30,31], purified by HPLC, and the TFA in the elution buffer exchanged with phosphate ions, and MS/MS analysis that was done at the University of Florida Biotechnology center showed 98% purity. His_6_-TMOF (1 mg) was dissolved in PBS buffer containing 0.5% CHAPS (1.0 mL) and 100 μL (100 μg) was incubated for 3 h at room temperature with solubilized gut membrane proteins (0.75 mL, 3.6 mg) in the presence of PBS 0.5% CHAPS pH 7.2 (100 μL) in a total volume of 0.95 mL. After incubation, BS^3^ (Bis[sulfosuccinimidyl]suberate) (0.950 mL), a cross linking agent (Thermo Fisher Scientific, USA), was added to a final concentration of 5 mM, and the reaction incubated for additional 30 min at 24 °C. The reaction was stopped by adding 75 μL Glycine (1 M) in PBS 0.5% CHAPS buffer pH 7.2 to a final concentration of 40 mM and the reaction was incubated for an additional 15 min at 24 °C. The crossed linked membrane proteins to His_6_-TMOF complex was purified by Ni affinity chromatography using a small column (5 mL) following manufacturer suggestions (Qiagen, USA). Briefly, the column was first equilibrated with wash buffer 1 (PBS, 0.15 M NaCl, 0.5% CHAPS buffer pH 7.2). After equilibration, the crossed linked gut membrane proteins were adsorbed unto the column and the column was then washed with wash buffer 1 (15 mL) followed with wash buffer 2 containing 20 mM imidazole (30 mL) and the cross linked His_6_-TMOF-membrane proteins complex eluted with wash buffer 3 (20 mL) containing 250 mM imidazole. Buffers 2 and 3 are similar to buffer 1 except that they contain 20 and 250 mM imidazole, respectively. Fractions (1 mL) were collected and the protein content in each fraction was assayed at 595 nm using a Bradford protein assay (Bio-Rad USA) (Figure 1A).

Fractions 51–56 and 58–61 were combined and dried using speed vac at 50 °C. The dried protein was then solubilized in SDS sample buffer 100 μL containing Tris-HCl, 2% SDS and 5% ME and 20% glycerol with tracking dye, heated at 90 °C for 5 min and centrifuged at 14,000 rpm to remove precipitated salts and proteins and that did not go back into solution. Samples (50 μL) were run using 10%-SDS PAGE and stained with Coomassie Brilliant blue [32]. The stained protein bands (M_r_ 55 kDa and M_r_ 45 kDa; Figure 1B) were excised digested with trypsin and analyzed by mass spectrometry (MS/MS) at the University of Florida Biotechnology Protein Core (http://www.biotech.ufl.edu/ProteinChem/ (accessed July 2007)). Gut extraction, cross linking to His_6_-TMOF, affinity chromatography and MS/MS analysis were repeated three times.

### 2.4. Cloning and Sequencing of AeaABC-TMOF Gut Receptor cDNA

MS/MS analysis of the SDS stained bands identified ATP-binding cassette transporter (EAT37643) in the genome of *Ae.aegypti* as a possible TMOF receptor. The genomic DNA was then analyzed for introns and exons using Lasergene Genomic Suite software (DNASTAR) and specific overlapping primers covering the length of the cDNA were synthesized (Table 1 and Figure 2A).

Primers were synthesized by Sigma Aldrich and were used to amplify by RT-PCR amplicons from the *Ae. aegypti* ABC-*tmf*A importer receptor transcript. RNA was extracted from adult female guts 72 h after the blood meal and amplicon sizes (nt), position (5’-3’) on the cDNA (see Figure 2A) and melting temperatures (*t_m_*) used by the different primer pairs for sequencing are shown.

Amplicons of 202–891 bp were amplified by RT-PCR reactions (20 μL) containing 4 μL 25 mM MgCl_2_, 2 μL 10 × PCR buffer (Applied Biosystems, Foster City CA), 6 μL sterile distilled water, 4 μL dNTP (10 mM of each dATP, dTTP, dCTP and dGTP), 1 μL RNase inhibitor (20 U), 1 μL MMLV reverse transcriptase (50 U) was prepared containing reverse primers (15 μM) each (Table 1) and *Ae. aegypti* gut RNA (1 μg). Reverse transcription (RT) was performed in a thermal cycler (Applied Biosystems) at 24 °C for 10 min, followed by 42 °C for 60 min, 52 °C for 30 min, 99 °C for 5 min, and 5 °C for 5 min. After RT, 3 μL 10× Buffer, 25.5 μL sterile distilled water, 2.5 U Amplitaq DNA polymerase (Applied Biosystems) and 15 μM of forward primers (Table 1) were added to the reaction mixture, PCR was carried out as follows: denaturation for 3 min at 95 °C (1 cycle), annealing for 4 min at 48 °C, extension for 40 min at 60 °C (1 cycle each), denaturation at 95 °C for 30 s, annealing for 30 s at 48 °C and extension for 2 min at 60 °C (40 cycles) with final extension for 15 min at 60 °C. Following PCRs, the cDNAs were separated by gel electrophoresis on 2% agarose gel in Tris-acetate-EDTA (TAE) buffer (pH 7.8) containing ethidium bromide at 100 V for 60 min. DNA bands corresponding to each amplicon nucleotides (nt) (Table 1) were visualized under UV lights, were cut from the gels, eluted with QIAquick gel extraction kit (Qiagen, Germantown, MD, USA) and cloned into TOPOpCR2.1 according to manufacturer instructions (Invitrogen, Carlsbad, CA, USA). INVaF’ *E. coli* cells were transformed, plasmids were purified with QIAprep Spin Miniprep kit (Qiagen), sequenced using BigDye Terminator v 3.1 Cycle Sequencing kit (Applied Biosystems) and analyzed at the University of Florida DNA sequencing core (http://langsat.biotech.ufl.edu/ (accessed June 2009)). A 5′RACE was used to sequence the 5′ end of the gene, a gene specific primer was synthesized (DB 2017, Table 1) and reverse transcription was performed with Superscript II reverse transcriptase [33]. After RT-PCR, 5 μL of the reaction mixture was removed and the dsDNA re-amplified using a second PCR mixture containing forward and reverse primers dT_17_ adapter and DB 2017, respectively (Table 1). To sequence the 3′ end of the cDNA primers DB 2016 and 3′ end reverse were used (Table 1). The overlapping sequences were joint together for a full-length cDNA and protein sequence using Lasergene Genomic Suite Software (DNASTAR) (Figure 2A,B and Figure 3) and deposited in the GenBank (Accession number MK895491).

### 2.5. Synthesis of AeaABC-TMOF Receptor dsRNA

Transcription vector pLITMUS28i (New England Biolabs) was used to synthesize dsRNA. A 528 bp amplicon (nucleotides 1712 to 2240, Figure 3) of *Aea*ABC-TMOF receptor was amplified using primers DB1083 (forward) 5′AACTATCGGGAGGTCAGAAACAACG3′ and DB1084 (reverse) 5′CGATGCTCCTACAACCATGGCTG3′ (Sigma-Aldrich), cloned into pLITMUS28i and amplified by PCR using T7 primer (5′ TAATACGACTCACTATAG 3′) (*t_m_* 41.4 °C). The dsRNA fragment carrying T7 promoter region at the 5′ end of the plus and minus strands was transcribed by T7 polymerase using HiScribe RNAi transcription kit (New England Biolabs, Ipswich, MA, USA). The transcribed dsRNA was precipitated in the presence of 5 M ammonium acetate (pH 5.2) and ethanol (100%), dissolved in sterile water and its concentration determined using DNA quant (Biochrom Ltd., Cambridge, UK).

### 2.6. Feeding Female Ae. aegypti with dsRNA

To find out if feeding *Aea*ABC-TMOF receptor dsRNA affects trypsin activity in the gut, dsRNA (8 μg/μL) in total volume of 20 μL containing 5% sucrose and 0.2% BSA was fed through a capillary for 7 days to three groups of female *Ae. aegypti* (80 per group). Controls were similarly fed without dsRNA. After 8 days, the female *Ae. aegypti* were fed blood on a live chicken and at intervals (8 and 34 h) after the blood meal females’ guts (5 guts/group) were dissected and analyzed for trypsin activity using BApNA [34]. 

### 2.7. Molecular Modeling of AeaABC-TMOF Receptor

Homology modeling of the *Aea* ABC TMOF receptor (Locus MK895491) transcript was run using the YASARA Structure program [35]. Eleven different models of *Ae. aegypti* ABC-TMOF receptor were built from the X-ray coordinates of the *Caenorhabditis elegans* multidrug transporter P-glycoprotein (P-gp) (PDB 4F4C), the mouse P-glycoprotein (PDB 4M1M), the mouse P-gp (PDB 4Q9L), and the mouse P-gp (PDB 5KO2 and 5KPI) [36,37,38,39]. A hybrid model of *Aea*ABC-TMOF receptor was built using the 11 models. PROCHECK [40] was used to assess the geometric quality of the three-dimensional model showing that all of the residues were correctly assigned in the allowed regions of the Ramachandran plot except for 2 residues (I1170 and K1253) which occur in the non-allowed region of the plot. ANOLEA [41] was then used to evaluate the model, showing that 47 residues of *Ae. aegypti* ABC-TMOF receptor out of 1307 residues exhibited an energy over the threshold value. These residues are located at the loop region connecting α-helices and β-sheets and the calculated QMEAN score for the *Aea*ABC-TMOF-receptor model is −1.72 [42]. 

The hydrophilic and hydrophobic regions distributed at the molecular surface of *Aea*ABC-TMOF receptor (hydrophobic surface colored orange, hydrophilic surfaces colored blue) were identified with the hydrophobicity surface option of Chimera [43]. The surface exposed electrostatic potentials of *Ae. aegypti* ABC-TMOF receptor were calculated and displayed (negatively and positively charged regions colored red and blue, respectively, and neutral regions colored grey) at the molecular surface with YASARA, using inner and outer dielectric constants that were applied to the proteins and the solvent, of 4.0 and 80.0, respectively. 

The decapeptide *Aea*TMOF (1YDPAPPPPPP10) was built as a left-handed α-helix using Chimera [43] and the structure was minimized using 1000 steps of steepest descent. Docking of TMOF to the two forked α-helical domains located at the top of the extracellular region of *Aea*ABC-TMOF receptor, was performed with YASARA [35]. Additional docking experiments used the GRAMM_X [44,45] and the ClusPro 2.0 [46,47,48] web servers. Molecular cartoons were drawn and rendered with the YASARA Structure and the UCSF Chimera packages [35,43]. 

### 2.8. Cloning and Expression of AeaABC-TMOF Receptor

Plasmid pTAC-MAT-Tag-2 a 5178 bp *E. coli* expression vector for cytoplasmic expression (Sigma-Aldrich, MO, USA) was used to clone and express ABC-TMOF-receptor protein in the bacterium inner membrane like SbmA, a bacterium inner membrane protein, that like ABC transporter facilitates movement of proline rich peptide into *E. coli* cytoplasm [49]. *Aea*ABC-TMOF receptor gene (Figure 3) with additional *Sma*I cloning sites at the 5′ and 3′ ends of the gene (3936 bp) was synthesized (GenScript, Piscataway, NJ, USA). The plasmid and the ABC-TMOF receptor gene were digested with *Sbm*I (5 units) following manufacturer recommendation (Thermo Fisher Scientific USA) and the gene inserted directionally and in frame into the multiple cloning site (Figure 4). 

The plasmid with the ligated gene was sequenced and analyzed by *Sma*I restriction enzymes analysis, cloned into competent *E. coli* cells (CGSC 8547, Section 2.1) using electroporation (BioRad, CA, USA). After electroporation, cells were spread on LB agar plates containing Kanamycin or Ampicillin and cells that were resistant to both antibiotics were selected on LB agar plates, grown in LB medium in the presence of Kanamycin and glycerol stock solutions prepared and stored at −80 °C. To express *Aea*ABC-TMOF-receptor, transformed *E. coli* cells were grown in LB medium in the presence of Ampicillin and Kanamycin and 0.4 mM isopropyl-1-thio-b-D-galactopyranoside (IPTG) at 37 °C until mid-log phase was achieved. The cells were then used to test binding of TMOF labeled with IPTG to its receptor.

### 2.9. AeaABC-TMOF Receptor Binding Assay and Kinetics

Mid-log phase bacteria harboring TMOF-receptor were harvested, diluted and 0.5 μL (8 × 10^8^ cells) were incubated in M9 medium (100 μL) pH 7.2 at 37 °C for 2 h in a shaker incubator in Eppendorf tubes with different concentrations of TMOF-FITC (0–0.4 μM) (specific activity = 25,396 fluorescence units/pmol) in the presence of TMOF (1 μmol) to reduce nonspecific binding of TMOF-FITC to the bacterial surface. After incubation, the cells were centrifuged at 14,000 rpm for 5 min at 4 °C, resuspended in PBS pH 7.2 (100 μL), vortexed, centrifuged, and the washing was repeated three times. After the last wash, the cells OD_600nm_ and fluorescence were read in GloMax multidetector system using a blue filter (excitation at 490 nm and emission at 510–570 nm). The amount of TMOF-FITC that bound the TMOF-receptor was determined from a linear calibration curve after plotting different concentrations of TMOF-FITC against fluorescence units. TMOF-FITC (H-YDPAPPPPPPK(FITC)K-OH) was synthesized at the University of Colorado Anschutz School of Medicine protein core. TMOF-FITC was purified by HPLC showing a single peak and mass spectrometry analysis of the peak identified the 3 expected ions at 455.7, 607.3 and 910.3 showing an expected M_r_ 1019.56 (Appendix A). After HPLC purification, the TFA ions were exchanged with phosphate ions. All data were corrected for nonspecific binding of TMOF-FITC (13%) to *E. coli* cells *SbmA^−^* containing an empty pTAC-MAT-2 plasmid and do not express the TMOF receptor. The results are expressed as pmol/OD_600_ so the binding to the receptor is per equal number of *E. coli* cells. The kinetics of TMOF-FITC binding to its receptor was followed at different concentrations of TMOF-FITC using 5 repetitions per concentration, the data was then fitted by nonlinear regression using GraphPad Prism and the K_D_, B_max_ and the affinity constant K_assoc_ determined. The binding experiments were repeated 3 independent times.

### 2.10. Transport of AeaTMOF-FITC into E. coli Cells Expressing AeaABC-TMOF Receptor in the Presence of Inhibitors

To find out whether the transport into *E. coli* cells expressing the TMOF receptor is proton driven or depends on ATP hydrolysis, *E. coli* cells (5 × 10^5^ cell) expressing the TMOF receptor were incubated in M9 medium (100 μL) with valinomysin (7 μM), DNP (27 μM), NaAzide (100 μM) and NaArsenate (20 mM) in the presence of *Aea*TMOF-FITC (0.4 μM) for 2 h at 37 °C. Control was run without the inhibitors. After incubation, the cells were washed 3× by centrifugation and the fluorescence was read after subtraction the initial background fluorescence of the medium and inhibitors without the *Aea*TMOF-FITC. The incubations were repeated 3 independent times and the results are expressed as means of 3 determinations ±SEM. 

### 2.11. Fluorescence Microscopy

Binding of TMOF-FITC to the *Aea*ABC-TMOF receptor and its transport into the recombinant *E. coli* cells were observed using Leica DM6B microscope with Leica DFC 7000 T camera and 63× magnification using oil immersion lens. Mid-log *E. coli* cells (10^8^ cells) expressing *Aea*ABC-TMOF-receptor were incubated with TMOF-FITC (see Section 2.6) in M9 medium (100 μL) overnight at 37 °C. After incubation, the cells were washed 3× in PBS pH 7.2 (as in Section 2.6) and aliquots (10 μL) were spread on a glass slide then covered with a cover slip and a drop of oil was applied to the top of the cover slip, and the cells observed by fluorescence microscopy. Cells without a TMOF-receptor containing an empty plasmid were used as control.

### 2.12. Statistical Analysis

Statistical analyses were determined using GraphPad Prism using two tailed Student’s *t* test and nonlinear regression. Results were considered significant when *p* < 0.05. Kinetic parameters K_D_ and B_max_ were determined from a nonlinear regression (*R^2^ >* 0.953) using GraphPad Prism. Each experimental point is a mean of 3–5 determinations ± SEM.

## 3. Results

### 3.1. Purification and Identification of AeaABC-TMOF Receptor

After affinity chromatography, SDS PAGE and MS/MS analysis of the two stained protein bands, 30 proteins were identified. Transmembrane analysis and protein function eliminated 26 of the initially identified proteins leaving four membrane associated protein genes from the *Ae. aegypti* genome: conserved hypothetical protein (EAT363580), transmembrane protein (EAT371176), transient receptor potential channel (EAT43552) and ATP-binding cassette transporter (EAT37643). Immunocytochemical-studies of *Aea*TMOF receptor using intact guts indicated that *Aea*TMOF binds to a receptor on the gut epithelial cells and is also found inside the cells [25], prompting us to sequence the cDNA of the ATP-binding cassette (ABC) transporter that may also be an importer like was shown in plants [29]. MS/MS analysis showed that *Aea-*ABC transporter is associated with the lower molecular weight (M_r_ 45 kDa) protein band resolved by SDS PAGE (Figure 1B).

### 3.2. Sequencing and Characterization of AeaABC-TMOF Receptor cDNA

Sequencing of the *Aea*ABC-TMOF receptor cDNA shows that the mature mRNA transcript has 3924 nt (accession number MK895491) and encodes a protein of 1307 amino acids (Figure 2A and Figure 3). 3′ RACE failed to identify the poly A tail that is probably too far downstream. A short (3924–3944) untranslated region (UTR) past the TGA at the 3′ end was amplified with a primer that was chosen from the *Ae. aegypti* genome sequence, whereas 5′RACE amplified a short upstream of UTR sequence (nt 1 to −12) (Figure 2A). The *Aea*ABC-*tmf*A receptor gene has 10 introns past the ATG start signal in the 18,470 bp DNA. One of the introns upstream is very long (12,130 bp) and a second downstream intron was much shorter (1385 bp). The other eight introns are much shorter between 61 and 91 bp (Figure 2B). The *Aea*ABC-*tmf*A receptor gene has 11 exons ranging from 172 to 713 bp with one exon 502 bp found after the TGA stop signal (Figure 2B). The protein sequence contains two ABC signature motifs LSGGQ at 572–576 and at 1206–1210 indicating that the *Aea*ABC-TMOF receptor uses ATP for active transport (Figure 3) (PDB: 2onj). A phylogenetic tree based on the homology of the amino acid sequences of several mosquito species using NCBI tree viewer and BLAST pairwise alignment (blast.ncbi.nlm.nih.gov) identified several mosquito species of *Anopheles, Aedes* and *Culex* that show 80–100% identity with the *Aea*TMOF-receptor (Figure 5).

### 3.3. Effect of AeaABC-TMOF Receptor dsRNA on Trypsin Activity in the Gut

Feeding female *Ae. aegypti* dsRNA (520 bp) caused significant inhibition of 38% (*p* = 0.012) and 43% (*p* = 0.0001) 8 and 34 h after the blood meal, respectively (Figure 6). These results indicate that *Ae. aegypti* ATP Binding Cassette transporter (EAT37643) that we identified is the TMOF receptor that regulates blood digestion and trypsin biosynthesis in *Ae. aegypti* gut after the blood meal.

### 3.4. Three-Dimensional Structure Analysis

The modeled *Aea*ABC-TMOF receptor consists of a typical V-shaped P-glycoprotein (Pgp) ATP binding cassette transporter structure, built from two asymmetrical *N*- and *C*- terminal ATP-binding domains (bd), *N*-ATPbd and *C*-ATPbd, located at the bottom of the structure, linked to an extended V-shaped α-helical domain (αHd). The α-Hd exhibits a hydrophobic region allowing the receptor to anchor in a membrane lipid bilayer (MLB), that splits the *Aea*ABC-TMOF receptor in the intracellular MLB domain into a V shaped structure that is linked to a smaller extracellular domain (Figure 7A). Both extracellular and intracellular domains located on both sides of the hydrophobic α-helical region, are predominantly hydrophilic (Figure 7B) and display electropositive and electronegative charged patches on their molecular surfaces (Figure 7C).

Docking experiments using *Aea*TMOF (YDPAPPPPPP) as a ligand, identified the peptide’s binding cavity at the fork of the V-shaped αHd structure (Figure 7D). The binding site of the *Aea*TMOF decapeptide exhibits both electropositive and electronegative charged surfaces (Figure 7E). In fact, both electrostatic interactions, hydrophilic and hydrophobic, stacking interactions and hydrogen bonds, anchor TMOF to the binding cavity (Figure 7G). The connectivity of α-helices and loops at the top of the extracellular region of the a-helical domain, is sufficiently flexible allowing TMOF to access and anchor to the binding site located in the intracellular region at the fork of αHd of the *Aea*ABC-TMOF receptor (Figure 7F). The interaction of *Aea*TMOF with the *Aea*ABC-TMOF receptor binding site (971FALGQIMPFMGYG983) show that an αhelical stretch is involved in the binding of *Aea*TMOF to the *Aea*ABC-TMOF receptor (Figure 8A,B). This sequence is found in the 45 kDa stained protein band that was purified by SDS PAGE and analyzed by MS/MS (Figure 1B and Appendix A).

### 3.5. Binding Kinetics of TMOF to AeaABC-TMOF Receptor

To find out the specificity of TMOF binding to *Aea*ABC-TMOF receptor, TMOF-FITC was incubated with *E. coli* cells *SbmA^−^* expressing *Aea*ABC-TMOF receptor in the inner bacterial cell membrane and compared with the binding of TMOF-FITC to *E. coli SbmA^−^* and *E. coli Sbm*A^+^ cells both contain empty pTAC-MAT2 plasmid. *E. coli* cells expressing the *Aea*ABC-TMOF receptor significantly bound more TMOF-FITC than *E. coli* SbmA^−^ and *E. coli Sbm*A^+^ (7.8-fold (*p* = 0.0018) and 3.7-fold (*p* = 0.0034), respectively) (Figure 9).

All our binding results, therefore, were corrected for nonspecific binding (13%) to the bacterial cell wall. Overnight incubations of TMOF-FITC with *E. coli* cells expressing *Aea*ABC-TMOF receptor and analysis of the bacterial cells by fluorescence microscopy show that the *Aea*ABC-TMOF receptor not only binds the TMOF-FITC but also imports it into the bacterial cells. *E. coli Sbm*A^−^ not expressing the receptor with an empty plasmid did not fluoresce. Short time incubations (2 h) of cells expressing the *Aea*ABC-TMOF receptor did not fluoresce similar to cells that did not express the receptor (Figure 10A,B).

As the *Aea*ABC-TMOF receptor is an importer all the kinetic binding studies were done for short time interval of 2 h. *Aea*ABC-TMOF receptor expressed in *E. coli Sbm*A^−^ cells and incubated with TMOF-FITC shows a concentration dependent specific binding with a K_D_ = 113.7 ± 0.018 nM + SEM and B_max_ = 28.7 ± 1.8 pmol/OD_600_ ± SEM (Figure 11, *n* = 5). From these results an affinity constant was calculated as K_assoc_ = 8.8 × 10^6^ M^−1^. Nonspecific binding (13%) was subtracted from each point.

### 3.6. TMOF Transport into E. coli Cells Expressing ABC-TMOF Receptor Is ATP Driven

Sequence analysis of ABC-TMOF receptor indicates that it encodes two ATP binding domains (Figure 7A). To determine whether the driving force that transports TMOF into the *E. coli* cells is ATP, we incubated *E. coli* cells expressing *Aea*ABC-TMOF receptor with *Aea*TMOF-FITC in the presence of DNP (27 μM), Valinomycin (7 μM), NaAzide (100 μM) and NaArsenate (20 mM). DNP and Valinomycin are ionophores that move protons across the inner membrane and affect the pH gradient and the membrane potential and do not depend on ATP, whereas NaAzide and NaArsenate inhibit ATPase activity blocking transport by ABC transporters that use ATP hydrolysis to import molecules into cells [27,50]. Our results show that both NaAzide and NaArsenate significantly inhibit the transport of *Aea*TMOF-FITC into the *E. coli* cells that express the *Aea*ABC-TMOF-receptor as compared with control cells (43%, *p* = 0.036 and 58%, *p* = 0.003, respectively). On the other hand, Valinomycin and DNP did not stop the transport of *Aea*TMOF-FITC into *E. coli* cells expressing *Aea*ABC-TMOF receptor as compared with control cells (*p* = 0.188 and *p* = 0.313, respectively) (Figure 12). These results indicate that *Aea*ABC-TMOF receptor imports TMOF-FITC into the bacterial cell by ATP hydrolysis.

## 4. Discussion

Earlier reports [22,23] showed that *Neobelleria Neb*TMOF (NPTNLH) and *Aedes aegypti Aea*TMOF (YDPAPPPPPP) stop the translation of the trypsin mRNA inside the gut epithelial cells indicating that the peptide is transported into the gut epithelial cells after binding a TMOF receptor. To purify the TMOF receptor we extracted female *Ae.aegypti* gut membranes 72 h after the blood meal [25]. In these guts the blood meal had been already digested and, therefore, proteins from the blood meal did not interfere with the isolation of the gut membrane proteins. To anchor potential gut receptor(s), a synthetic (His)_6_TMOF was incubated and cross linked to the extracted soluble membrane proteins, and the TMOF–gut membrane protein complex purified by Ni affinity chromatography followed by SDS PAGE. Two protein bands were detected (Figure 1A,B), extracted and analyzed by MS/MS. Four membrane genes were identified in the *Aedes aegypti* genome (accession number GCA_002204515.1), however, only one of the genes ATP Binding Cassette transporter (ABC-transporter) could function as a receptor and importer.

ABC proteins are part of a transporter superfamily with P-loop motif [51,52,53,54,55,56] but they are also involved in many other biochemical and physiological processes. They play a role in the resistance to different Bt toxins by reducing the binding of Cry toxins to the brush border membrane vesicles in different lepidopteran species [57,58,59]. Thus, the ABC transporters have important role in xenobiotic detoxification and Bt-resistance. Although there are many published reports on the role of ABC as importers in bacteria and in plants no report has been published on ABC importers in insects [29,60]. The ABC transporter AtABCB14 in plants is a malate importer and modulates stomatal response to CO_2_. Sequencing the ABC transporter cDNA show that the cDNA is 3924 bp with two ABC signature motifs (LSGGQ) indicating that ATP is involved in the transport. ABC genome has 10 introns and 11 exons (Figure 2). Blast of the sequence to find out similar transcripts with 80–100% identity confirmed that several *Aedes*, *Anopheles* and *Culex* species have a similar receptor (Figure 5). Indeed, TMOF has been shown to affect these mosquitoes by controlling their trypsin biosynthesis [31] indicating that in these mosquitoes trypsin biosynthesis is also regulated using ABC-TMOF receptors.

Feeding female *Ae. aegypti* ABC-TMOF receptor dsRNA inhibited the trypsin biosynthesis in the gut at 8 and 34 h after the blood meal (Figure 6). These results indicate that the ABC-TMOF receptor is involved in the regulation of trypsin biosynthesis in the gut and knocking down the ABC-TMOF receptor reduced the amount of trypsin synthesized in female *Ae. aegypti* midgut. We did not feed *gfp* dsRNA as a second control because we have reported [61] before that feeding or injected *gfp* to female or larvae *Ae. aegypti* does not affect egg development or blood digestion. Similar observations on the use of dsRNA to inhibit receptors were reported for steroid receptors in *Ae. aegypti*, sex peptide receptor in *Drosophila*, and vitellogenin receptor in ticks that caused reduction in JH signaling in *Ae. aegypti*, oviposition rate in *Drosophila* and egg development in ticks, respectively [62,63,64].

The three-dimensional modeling of the *Aea*ABC-TMOF receptor identified a binding site for *Aea*TMOF (Figure 7D) showing that the hormone can penetrate the inner membrane bilayer through an opening at the space between the outer and inner membrane bilayer (Figure 7F) and specifically interact using hydrogen bonding as well as hydrophobic and hydrophilic bonding with a helical stretch (Figure 8A,B).

SbmA is a bacterial inner membrane protein that forms a dimer and is involved in the import of peptides, proline rich peptides, nucleic acids, antisense peptides, and several oligomers into the *E. coli* cytoplasm. SbmA 3D conformation resembles *Aea*ABC-TMOF transporter lacking the nucleotide binding domain (Figure 7A) [26]. As *E. coli* S*bmA^+^* has shown to import proline rich peptides like Bac7(1-35) [26], the *E. coli* cells that we used to clone and express the *Aea*ABC-TMOF receptor lacked this importer and are *SbmA*^−^. When *E. coli SbmA*^−^ and *E. coli SbmA^+^* cells were incubated in the presence of *Aea*TMOF-FITC and compared with *E. coli SbmA*^−^ expressing *Aea*ABC-TMOF receptor, 31 pmol of TMOF-FITC bound the cells expressing the receptor whereas *E. coli* cells *Sbm*A^−^ and *SbmA^+^* bound 4 and 8.4 pmol of *Aea*TMOF-FITC, respectively, indicating that SbmA is not an efficient transporter of *Aea*TMOF as compared with *Aea*ABC-TMOF-receptor. As *Aea*TMOF stops the translation of mosquito’s gut trypsin mRNA [23,25] the hormone has to enter the epithelial cells cytosol and affect either the ribosomes or the trypsin mRNA. Incubating *E. coli* cells *SbmA*^−^ expressing *Aea*ABC-TMOF receptor and *E. coli SbmA^−^* cells overnight in the presence of *Aea*TMOF-FITC followed by examining the cells under a fluorescence microscope show that only bacterial cells expressing the *Aea*ABC-TMOF receptor fluoresce (Figure 10a,b). These results and earlier cytoimmunochemical studies of the binding and the transport of *Aea*TMOF into mosquito gut epithelial cells [25] confirm that the *Aea*ABC-receptor functions as an importer similar with ABC importers expressed in plants [29].

The binding affinity of TMOF to its *Aea*ABC-TMOF receptor shows high affinity (K_D_ = 113.7 ± 18 nM ± SEM, B_max_ = 28.7 ± 1.8 pmol + SEM, and K_assoc_ = 8.8 × 10^6^ M^−1^) (Figure 11). These results agree with earlier studies that were done using *Ae. aegypti* gut membrane preparations (K_D_ = 460 ± 70 nM ± SEM, B_max_ = 0.1 pmol/gut and K_assoc_ = 2.2 × 10^6^ M^−1^) [25]. Our earlier studies also characterized a low affinity receptor [25] whereas in this study the affinity chromatography and SDS PAGE procedure that we used aimed to purify a high affinity TMOF receptor. It is possible that a second low affinity receptor [25] did not bind the column or was eluted with the low imidazole fraction and we missed it. In future studies we will try to find out if a low affinity receptor can be also purified, cloned and sequenced. The transport of *Aea*TMOF into the recombinant *E. coli* cells does not depend on electrochemical transmembrane gradient but on ATP hydrolysis (Figure 12).

In conclusion, this report shows for the first time that a unique *Aea*ABC-TMOF-receptor binds TMOF with high affinity and imports *Aea*TMOF into bacterial cells expressing the receptor. These observations allow us to hypothesize how *Aea*TMOF controls trypsin biosynthesis in the mosquito’s gut. After the release of *Aea*TMOF from the mosquito ovary into the hemolymph the hormone binds *Aea*ABC-TMOF receptor on the gut epithelial cells with high affinity, and is imported into the epithelial cell cytosol subsequently stopping the translation of the trypsin message by either affecting the ribosomes directly like the proline rich oncocin that binds to the 70S ribosome of *Thermus thermophilus* at the peptide exit tunnel and prevents protein translation [65], or alternatively affecting the trypsin transcript. More work is needed to find the final steps that control blood digestion in such an important disease transmitting vector.

## Figures and Tables

**Figure 1 biomolecules-11-00934-f001:**
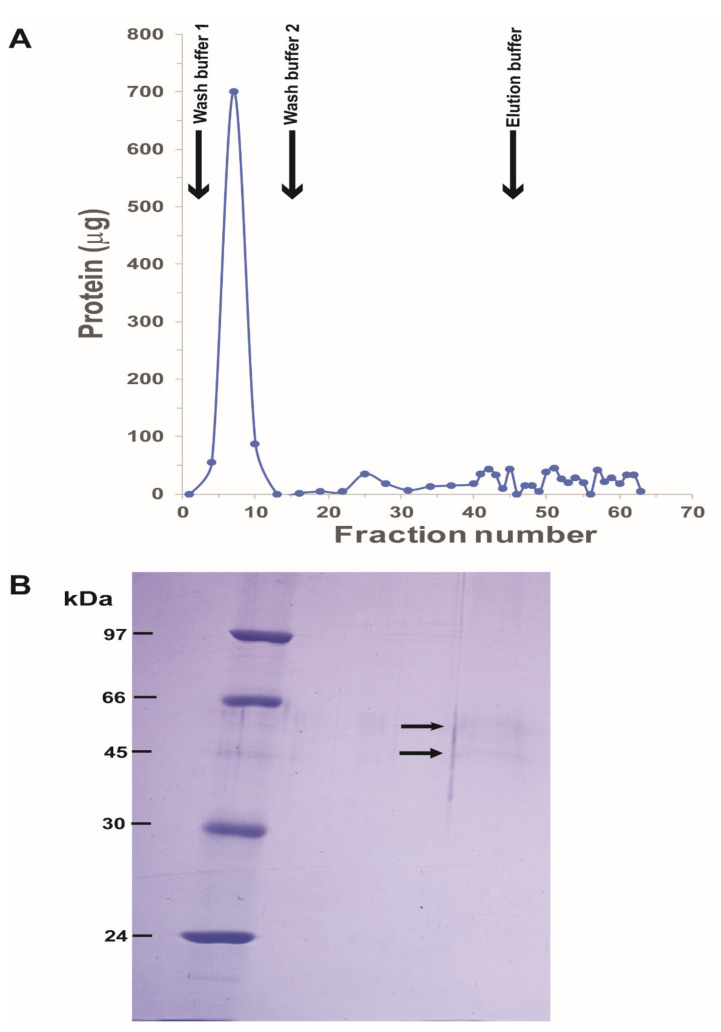
(**A**) Purification of His_6_-TMOF cross linked to solubilized *Ae. aegypti* TMOF receptor by Ni affinity chromatography. The cross-linked proteins to His_6_-TMOF were adsorbed to a Ni affinity column, the column was washed with buffer 1 (PBS, 0.15 M NaCl, 0.5% CHAPS) (15 mL) followed with wash buffer 2 (wash buffer 1 with 20 mM imidazole) (30 mL) and the crossed linked proteins to TMOF complex were eluted with 20 mL of elution buffer (buffer 1 containing 250 mM imidazole). Fractions (1 mL) were collected and analyzed for protein. Fractions (51–56 and 58–61) were collected and dried by speed vac and rehydrated with SDS sample buffer. The purification procedure was repeated three times. (**B**) SDS PAGE of the rehydrated His_6_-TMOF cross linked proteins after the Ni affinity chromatography. Protein standards (M_r_ 97 kDa to 24 kDa) were run in the left lane and TMOF cross linked membrane proteins complex in the right lane. After staining the gel, two faint protein bands were detected M_r_ 55 kDa and M_r_ 45 kDa (arrows). The bands were cut and analyzed by MS/MS. The SDS-PAGE was repeated three times showing the same faint bands.

**Figure 2 biomolecules-11-00934-f002:**
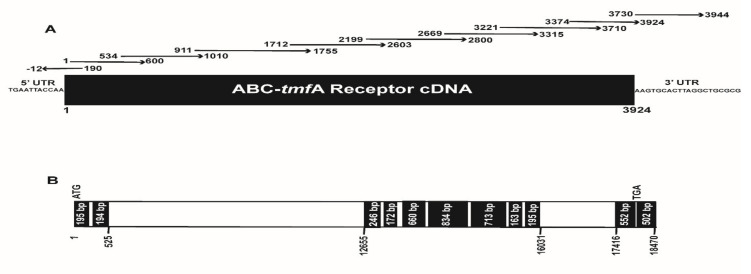
(**A**) *Aea*ABC-*tmf*A receptor cDNA map and sequencing strategy. Arrows show the size of the amplicons that were amplified by PCR, cloned and sequenced. The over lapping amplicons allowed a reliable assembly of the cDNA. The cDNA is 3924 bp long and the 5′ and 3′ UTRs are also shown. (**B**) A map of the *ABC-tmf*A-importer gene showing 11 exons (black color) and 10 introns (white color). The gene is 18,470 bp long starting at the ATG and with 502 bp past the TGA stop signal.

**Figure 3 biomolecules-11-00934-f003:**
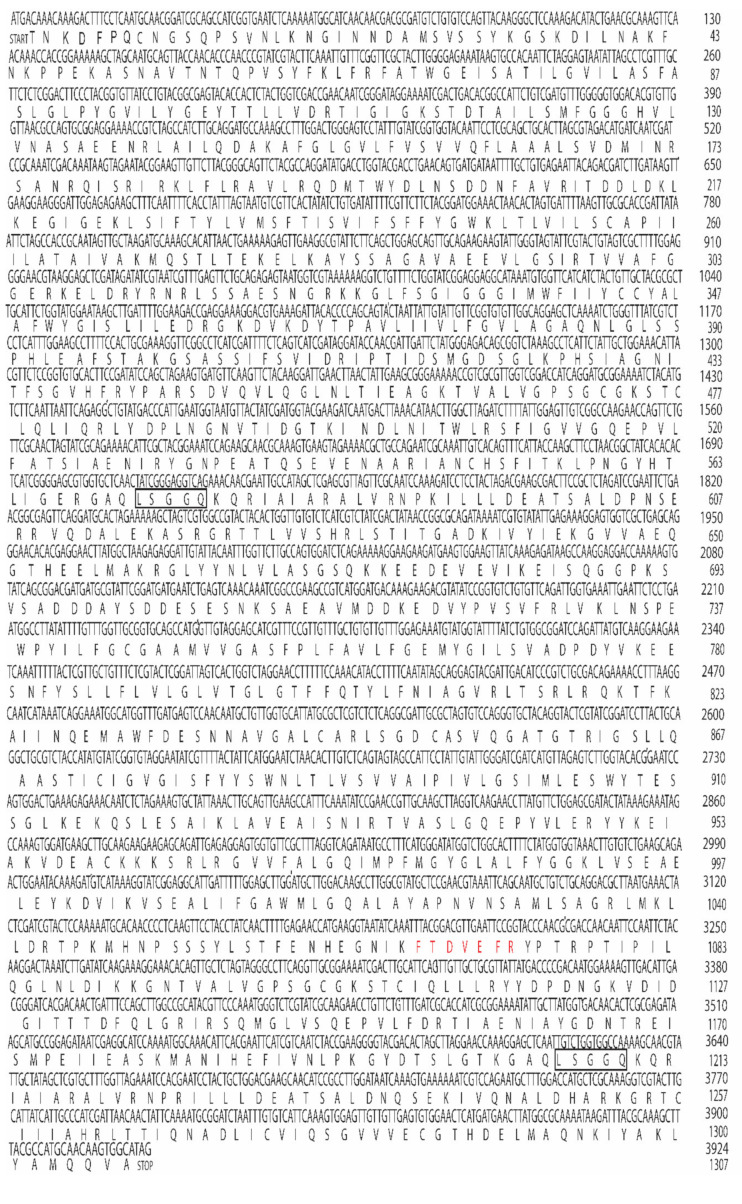
*Aea*ABC-receptor nucleotides and translated amino acids sequences. The translated protein has 1307 amino acids and two ATP signature motifs (LSGGQ) shown in black square boxes.

**Figure 4 biomolecules-11-00934-f004:**
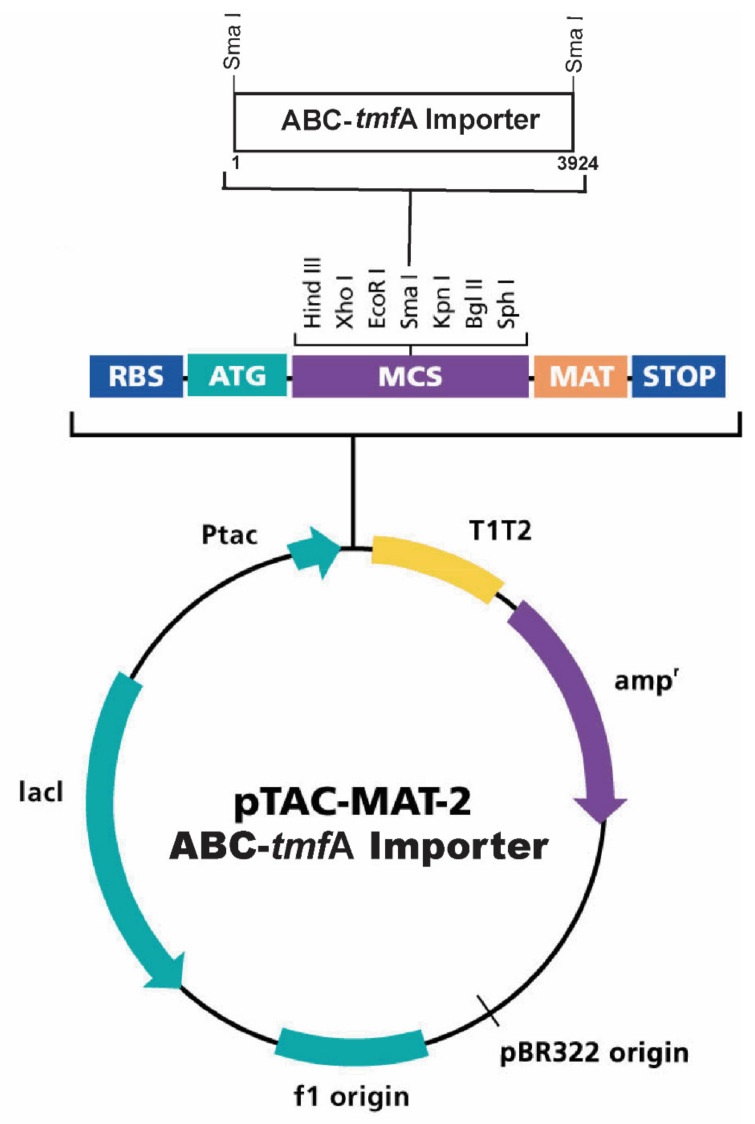
A graphic map of plasmid pTAC-MAT-2 that was used to directionally clone the *Aea*ABC-*tmf*A receptor gene at the *Sma*I cloning site. The plasmid is Amp^R^ with a *lac*I gene allowing the use of IPTG to express the recombinant AeaABC-TMOF receptor in the inner membrane of *E. coli*.

**Figure 5 biomolecules-11-00934-f005:**
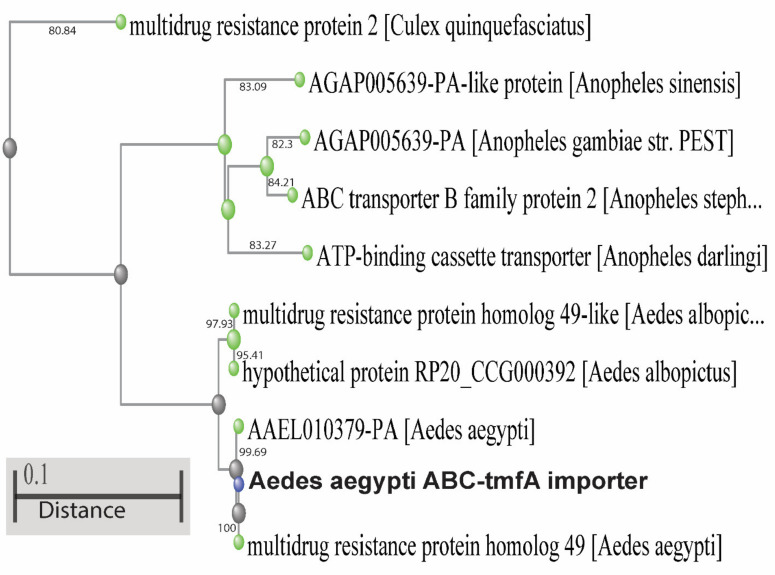
A phylogenetic tree of *Aea*ABC-TMOF receptor using sequence homology (80–100%). Several sequences from *Ae. aegypti*, *Ae. albopictus*, *Anoheles darlingi, Anopheles stephensi, Anopheles gambiae, Anopheles sinensis* and *Culex quinquefasciatus* show sequence similarities to the *Aea*ABC-TMOF receptor. Using Blast pairwise alignments (blast.ncbi.nlm.nih.gov) the distance between the sequences is shown as a bar on the bottom left corner.

**Figure 6 biomolecules-11-00934-f006:**
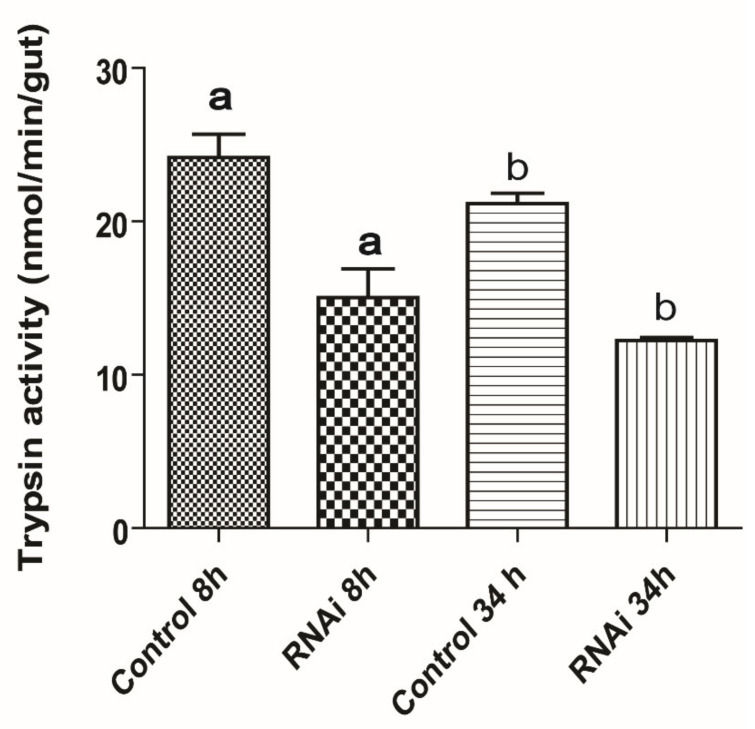
The effect of feeding *Aea*ABC-TMOF receptor dsRNA to female mosquitoes. Groups of female *Ae. aegypti* (five female/group) were fed on 5% sucrose and 0.2% BSA solution containing dsRNA (8 μg/μL) in capillary tubes for 7 days. Control groups were fed the same diet without dsRNA. After 7 days the female mosquitoes were fed blood on a chicken and at 8 and 34 h after the blood meal the guts were analyzed for trypsin activity using BApNA [34]. At 8 and 34 h after the blood meal Trypsin activity was significantly inhibited 38% and 43%, respectively, as compared with control groups. Results are expressed as means of three determination ± SEM and significant differences between control and dsRNA fed was determined using two paired Student’s *t* test. dsRNA of *gfp* fed to females does not have an effect on blood digestion or egg development. ^a,b^ Significant inhibitions at 8 h (*p* = 0.012) and 34 h (*p* = 0.0001).

**Figure 7 biomolecules-11-00934-f007:**
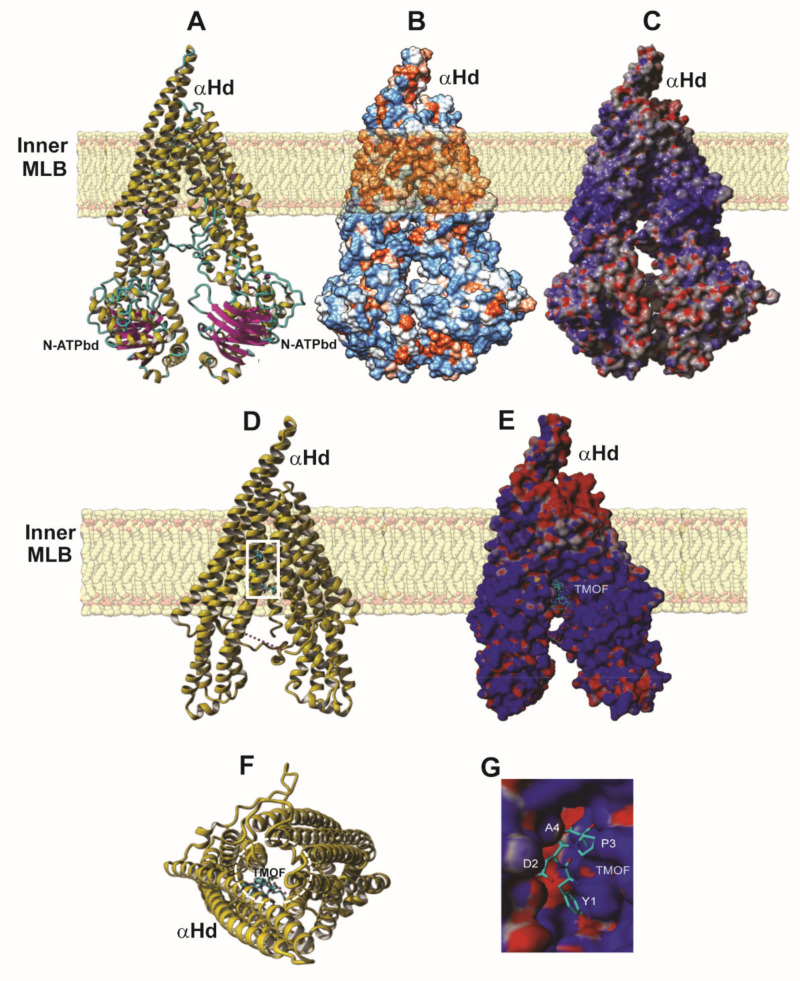
Three-dimension molecular model presentation of *Aea*ABC-TMOF receptor. (**A**) Ribbon diagram of the V-shaped three-dimensional model of *Aea*ABC-TMOF receptor. The intracellular ATP-binding domains (ATPbd) located at the bottom of the model are associated with the intracellular α-helical domain (αHd), that partly protrudes out of the cell surface. (**B**) Hydrophilic (colored blue) and hydrophobic (colored orange) patches distributed on the molecular surface of *Aea*ABC-TMOF receptor. The hydrophobicity of the α-helices allows the receptor to embed in the membrane lipid bilayer (MLB). (**C**) Surface electrostatic potential regions distributed on the molecular surface of the *Aea*ABC-TMOF receptor that are electronegatively and electropositively charged are colored red and blue, respectively. Neutral regions are colored grey. (**D**) Ribbon diagram showing *Aea*TMOF (colored green) docked into the binding site located at the fork of αhelical binding domain (αHbd). Dotted pink line shown in D and F, indicates that in these graphical presentations of the *Aea*ABC-TMOF receptor the N-ATPbd of the molecule is missing. (**E**) Surface exposed electrostatic potentials of the αHd showing a single docking position of TMOF (colored cyan) at its binding site. Electronegative and electropositive charged regions are colored red and blue, respectively, and the neutral regions are colored grey. (**F**) Upper front view of the αHd containing a single docking position of TMOF (colored cyan), showing the open space across the α-helices and loops (dashed white open circle) at the top of the αHd that allows TMOF to enter and bind inside the binding cavity. (**G**) Enlargement of (E) showing the binding of TMOF (colored cyan) in the binding cavity in the αHd of *Aea*ABC-TMOF receptor. Several of the amino acids of TMOF are in white letters.

**Figure 8 biomolecules-11-00934-f008:**
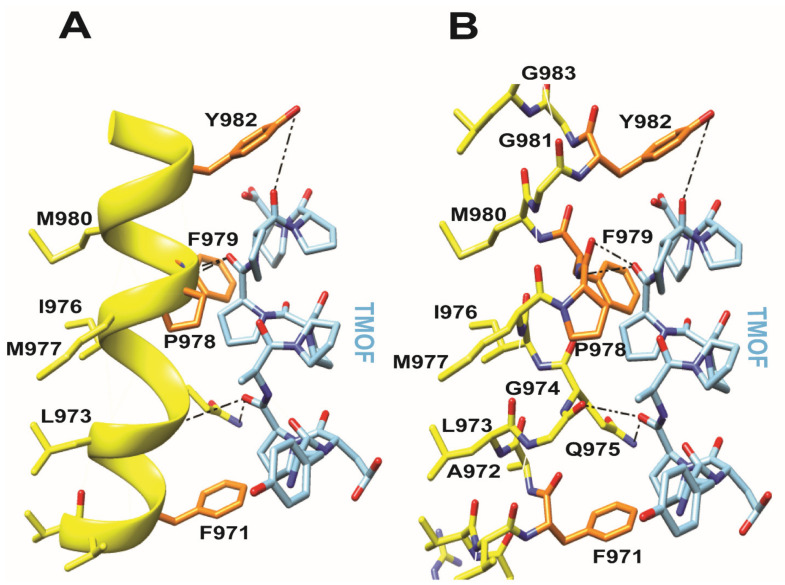
The interaction of *Aea*TMOF with the *Aea*ABC-TMOF receptor binding site sequence (971FALGQIMPFMGYG983) is shown in (**A**) as helical presentation (yellow color) and in (**B**) as a wireframe presentation, indicating that the binding of AeaTMOF (blue color) to its receptor involves hydrogen bonding (broken lines), hydrophobic and hydrophilic interactions.

**Figure 9 biomolecules-11-00934-f009:**
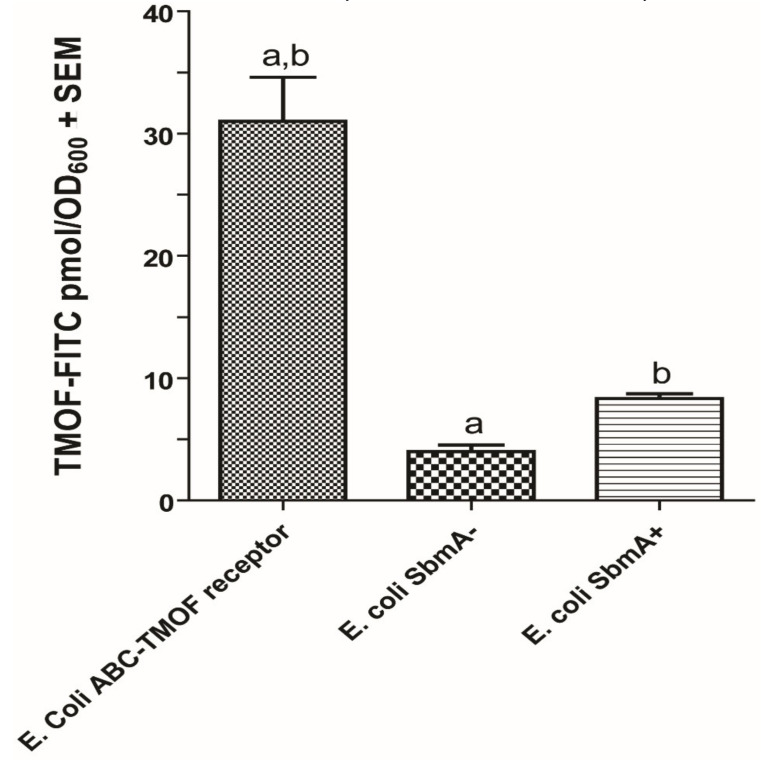
Binding of *Aea*TMOF-FITC to recombinant *E. coli SbmA^−^* cells (8 × 10^8^ cells) expressing *Aea*ABC-TMOF receptor (left bar) in comparison to non-specific binding to *E. coli SbmA^−^* cells (middle bar) and low binding to *E. coli SbmA^+^* cells (right bar). The bacterial cells were incubated with *Aea*TMOF for 2 h at 37 °C in M9 medium. The results are expressed as means of three determinations ± SEM. Significant differences were determined using two tailed Student’s *t* test. ^a,b^ Significant differences (*p* = 0.0018 and *p* = 0.0034, respectively).

**Figure 10 biomolecules-11-00934-f010:**
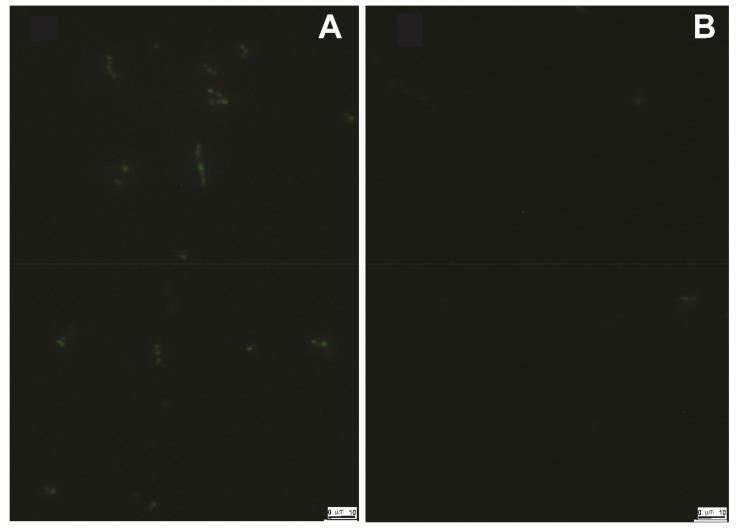
Fluorescence microscopy images of recombinant *E. coli* cells expressing *Aea*ABC-TMOF receptor (**A**) and *E. coli* cells with an empty pTAC-MAT-2 plasmid (**B**). *E. coli* cells (10^8^ cells) were incubated with *Aea*TMOF-FITC overnight in M8 medium at 37 °C. After incubation, the cells were washed 3X and observed by fluorescence microscopy. *E. coli* cell expressing the *Aea*ABC-TMOF receptor highly fluoresce (**A**) whereas bacterial cells not expressing the *Aea*ABC-TMOF receptor barely fluoresce (**B**). Bars are equal to 10 μm.

**Figure 11 biomolecules-11-00934-f011:**
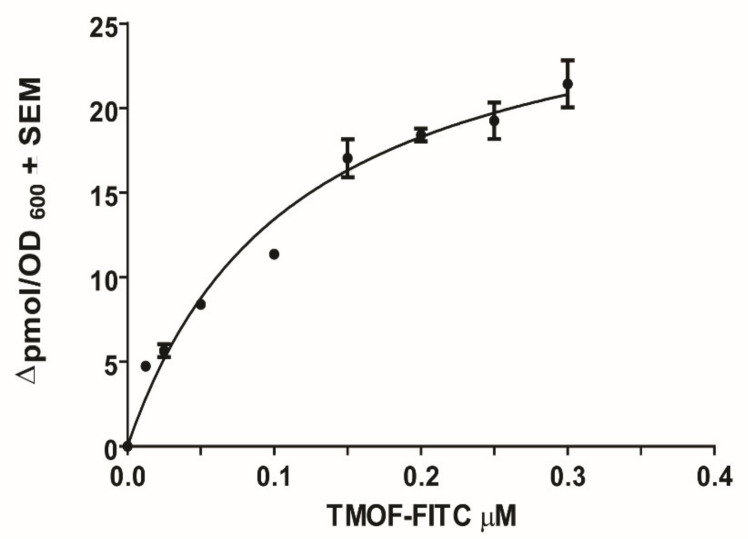
Specific binding of *Aea*TMOF-FITC to *E. coli* cells expressing AeaABC-TMOF receptor showing Michaelis Menten binding kinetics using a nonlinear regression (R^2^ = 0.9526). Binding results were corrected for nonspecific binding (13%) to the bacterial cell wall. Each point is a mean of five determinations ± SEM. The data represent one experiment of three independent experiments with similar results. K_D_ = 113.7 ± 18 nM ± SEM, B_max_ = 28.7 ± 1.8 pmo/OD_600_ ± SEM and K_assoc_ = 8.8 × 10^6^ M^−1^.

**Figure 12 biomolecules-11-00934-f012:**
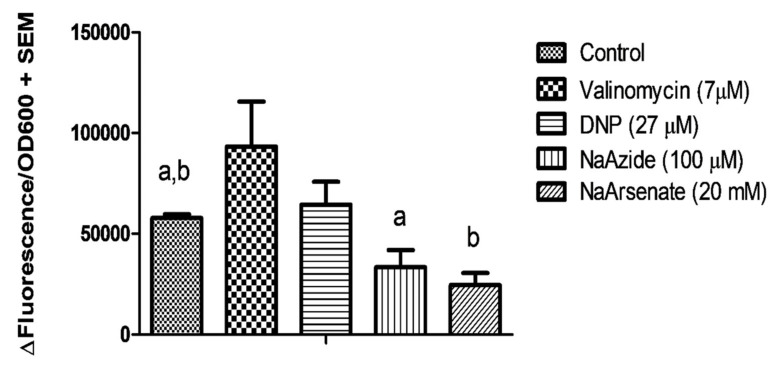
TMOF transport into *E. coli* cell expressing *Aea*ABC-TMOF-receptor is ATP driven. *E. coli* cells expressing TMOF receptor were incubated with Valinomycin (7 μM), DNP (27 μM), NaAzide (100 μM) and NaArsenate (20 mM) in the presence of *Aea*TMOF-FITC for 2 h and the transport into the bacterial cells was followed by fluorescence determination. Significant inhibition of the *Aea*TMOF-FITC transport into the bacterial cells of 43% and 58% was observed with NaAzide and NaArsenate, respectively. ^a,b^ Significant difference from control (*p* = 0.036 and *p* = 0.003, respectively).

**Table 1 biomolecules-11-00934-t001:** Primers used for sequencing ABC *tmf*A importer cDNA from *Ae.aegypti* gut.

Primers	Primer Sequence (5’-3’)	Position (5’-3’)	Amplicon (nt)	*tm (C)*
DB 2000 (forward)	ATGACCAAACAAAGACTTTCCTCA			64
DB 2001(reverse)	CAGGTCGTACCAGGTCATATCCTGG	1–600	600	69
DB 2002 (forward	AATAAGTAGAATACGGAAGTTGTTC			60
DB 2003 (reverse)	CCAGAATGCAAGCGCGTAGCAACAGT	534–1010	516	72
DB 2004 (forward)	GGGAACGTAAGGAGCTCGATAGAT			66
DB 2005 (reverse)	ATTTTTGGATTGCGAACTAACGCTC	911–1755	864	66
DB 2006 (forward)	AACTATCGGGAGGTCAGAAACAACG			67
DB 2007 (reverse)	CGATGCTCCTACAACCATGGCT	1712–2603	891	68
DB 2008 (forward)	GAATTCTCCTGAATGGCCTTATATTT			63
DB 2009 (reverse)	GGATATTTGAAATGGCTTCAACTGCA	2199–2800	601	66
DB 2010 (forward)	CAATAGTAGCCATTCCTATTGTATTGG			63
DB 2011 (reverse)	CCGCAACCTGAAGGCCCTACTAGA	2669–3315	646	70
DB 2012 (forward)	ACCCAACGCGACCAACAATTCCAAT			70
DB 2013 (reverse)	TCCAAGGCGGATGTTGCTTCG	3221–3710	489	68
DB 2014 (forward)	ACATTGACGGGATCACGACAACTG			68
DB 2015 (reverse)	CTATGCCACTTGTTGCATGGAGTAAA	3374–3924	550	67
*3’ End*				
DB 2016 (forward)	GTCCAGAATGCTTTGGACCCAT			57
3’End (reverse)	CGCGCAGCCTAAG	3730–3944	214	60
*5’ RACE:*				
dT17 adapter (forward)	GACTCGAGTCGACATCGA(T)17			74
DB 2017 (reverse)	TGAAGTACGATACGGGTTGGG	12–190	202	57

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
