# Peer review of "Cloning and Characterization of Aedes aegypti Trypsin Modulating Oostatic Factor (TMOF) Gut Receptor"

_biomolecules, 2021, doi:10.3390/biom11070934_

Round 1

Reviewer 1 Report

D. Borovsky reported the existence of an oostatic factor alfeady back in 1988. Initially, the finding met with scepticism, but it turned out to be correct. The final proof of its mode of action took until to date with the presentation of the descrption of the TMOF receptor. The identification of the structure of the receptor proved to be much more difficult than the ones of most GPCRs. Many different methods were needed.

In my opinion this is a truly outstanding piece of work. I checked the manuscript for the arguments in favor of the functionality of the receptor. The reported data are in agreement with what published before on the supposed mode action of TMOF: I am convinced that this is indeed the TMOF receptor of Aedes. I agree with what the authors write on line 578- "This report shows for the first time that a unique AeaABC-TMOF receptor binds TMOF with high affinity and imports  AeaTMOF into bacterial cells expressing the receptor."

The paper is very well well written, only a few typing errors:

line 72  quin..

lines 97+98 E. coli in titalic

line 363   contains

line 367 insect names in italic

line 464: the fluorescence microscopy images were for some technical reason not visible on my computer

line 539  Ae.aegypti in italic

Line 548 penetrate

Reference list: bring uniformity in the layout

Author Response

I would like to thank the reviewer for his suggestions and followed his suggestions to the letter.  Line 72 Quinqefasciatus has been changed to quinquefasciatus.  The q is higlighted in red color

Lines 97 and 98 E. coli has been itelized and they are highlighted in red color.

line 367 the insect names are all itelized and they are highlighted in red.

line 464 the fluorescence image is visible on my computer this has to be a glitch on the reviewer's computer.

line 539 there is no Ae. aegypti on this line, however, there is Ae. aegypti on line 540 but it is itelized.  

Line 548 the word penetrated has been changed to penetrate

All the references have been now formatted and they are uniform.

Reviewer 2 Report

The manuscript from Borovsky et al. gives a detailed description of the identification of the TMOF receptor from Ae. aegypti, showing it is an ABC transporter. The transporter is cloned and expressed, and the authors modeled TMOF binding sites and showed functional activity of the expressed transporter. I find the study solid and interesting that would benefit the readers of Biomolecules.

Here are a few minor suggestions for the authors to consider:  

  1. To make the article more appealing to a larger audience, I suggest the authors consider adding a few sentences in the abstract/introduction session, pointing out the significance and broader implication of this study. Although the authors did mention some in the discussion part, I think highlighting these in the beginning would be helpful.
  2. In section 3.5, the binding kinetics of TMOF to the transporter is determined with 2 hours of incubation. The import of TMOP into the cells is determined with overnight incubation. Since the authors claim the 2hours incubation is for measuring the binding, I assume the authors did not see the accumulation of TMOP in the cell with 2 hours of incubation then? The authors implied it in the text. To make it more apparent to the readers, please consider showing the control or specify it in the text.
  3. The high-affinity binding site identified in this study agrees with the authors’ previous work, as mentioned in line 574. In addition, in the previous study, a low-affinity site is also reported, and they also seem to contribute significantly to the number of TMOF binding sites in the gut. Could the authors give a few comments on this low-affinity site? Do the authors think a different unknown type of receptor is involved and unidentified?
  4. In line 353, I assume 3924 nt mRNA code 1308 amino acids, instead of 3924 amino acids as written in the text?
  5. I think the overall grammar of the paper could be improved. In many places, what should be multiple sentences were written into one, etc.

Author Response

All the suggested modification by this reviewer were taken into consideration and acted upon. 

  1. To make the article more appealing to a larger readership I have inserted a paragraph at the end of the introduction see lines 84 to 88.
  2. To answer the question why we used over night incubation of the transformed E. coli cells with  TMOF-FITC and not 2 h incubations I added a sentence that after 2 h incubations the  TMOF expressed receptor in transgenic E. coli cells did no fluoresce and they looked like the control photograph in Fig. 10 b.  See lines 467-468.
  3. A comment on the low affinity receptor and why we were not able to get it out is given in lines 586-591.
  4. I corrected the length of the receptor as was suggested as 1307,  The reviewer suggested 1308 however, the TGA at the end of the transcript is not converted into an amino acid only the first ATG is converted into Methionine.  See line 358.
  5. I followed the reviewer suggestions and combed the text for too short sentences that were combined to give a better readable text. see line 230, line 248, line 449 and line 527.

I would like to thank this reviewer for his suggestions to improve the manuscript and I hope that he accepts our revisions to the manuscript.  I am attaching a final version of the manuscript with the changes that reviewer 1 and 2 suggested.
